# Case management interventions in chronic disease reduce anxiety and depressive symptoms: A systematic review and meta-analysis

**Angelika D. Geerlings**[1], **Jules M. Janssen Daalen**[1], **Jan H. L. Ypinga**[1], **Bastiaan R. Bloem**[1], **Marjan J. Meinders**[2], **Marten Munneke**[1], **Sirwan K. L. Darweesh**[1]*

1 Department of Neurology, Donders Institute for Brain, Cognition and Behaviour, Center of Expertise for Parkinson & Movement Disorders, Radboud University Medical Center, Nijmegen, The Netherlands, 2 Scientific Center for Quality of Healthcare, Radboud Institute for Health Sciences, Radboud University Medical Center, Nijmegen, The Netherlands

* sirwan.darweesh@radboudumc.nl

**Data Availability Statement:** All relevant data are within the paper and its Supporting Information files.

## Abstract

### Background

There is no systematic insight into the effect of case management on common complications of chronic diseases, including depressive symptoms and symptoms of anxiety. This is a significant knowledge gap, given that people with a chronic disease such as Parkinson Disease or Alzheimer's Disease have identified care coordination as one of their highest priorities. Furthermore, it remains unclear whether the putative beneficial effects of case management would vary by crucial patient characteristics, such as their age, gender, or disease characteristics. Such insights would shift from "one size fits all" healthcare resource allocation to personalized medicine.

### Objective

We systematically examined the effectiveness of case management interventions on two common complications associated PD and other chronic health conditions: Depressive symptoms and symptoms of anxiety.

### Methods

We identified studies published until November 2022 from PubMed and Embase databases using predefined inclusion criteria. For each study, data were extracted independently by two researchers. First, descriptive and qualitative analyses of all included studies were performed, followed by random-effects meta-analyses to assess the impact of case management interventions on anxiety and depressive symptoms. Second, meta-regression was performed to analyze potential modifying effects of demographic characteristics, disease characteristics and case management components.

**Funding:** The authors received no specific funding for this work. The funders had no role in study design, data collection and analysis, decision to publish, or preparation of the manuscript.

**Competing interests:** The authors have declared that no competing interests exists.

## Results

23 randomized controlled trials and four non-randomized studies reported data on the effect of case management on symptoms of anxiety (8 studies) or depressive symptoms (26 studies). Across meta-analyses, we observed a statistically significant effect of case management on reducing symptoms of anxiety (Standardized Mean Difference [SMD] = - 0.47; 95% confidence interval [CI]: -0.69, -0.32) and depressive symptoms (SMD = - 0.48; CI: -0.71, -0.25). We found large heterogeneity in effect estimates across studies, but this was not explained by patient population or intervention characteristics.

## Conclusions

Among people with chronic health conditions, case management has beneficial effects on symptoms of depressive symptoms and symptoms of anxiety. Currently, research on case management interventions are rare. Future studies should assess the utility of case management for potentially preventative and common complications, focusing on the optimal content, frequency, and intensity of case management.

## 1. Introduction

The increasing prevalence of chronic disease poses a substantial burden to the capacity of health care services to provide adequate care the treatment requires continuous monitoring and ongoing interdisciplinary collaboration between health care providers of various disciplines, who ideally deliver proactive care [1]. However, current health care systems are typically designed to treat chronic diseases using a "one size fits all" approach instead of tailoring care to each patient's individual needs [2, 3]. Consequently, persons with chronic disease often become responsible for coordinating their care. As a result, patients receive fragmented and ineffective care in achieving the desired health outcomes [1, 3]. This increases health care costs and causes an unnecessary burden on patients and their carers, which, in turn, negatively affects their quality of life.

To address these challenges, case management has been introduced to improve the care coordination [4, 5]. Field defines case management in different ways [4, 6]. However, the everyday basis for each definition is that case management is a collaborative process involving one case manager or a small team that plans, coordinates and reviews the delivery of health care services to meet a patient's individual needs [7]. According to this integrated care approach, a case manager takes over the responsibility of managing non-acute services. The interpretation of case management can vary substantially. Still, common core elements include the development and review of individualized care plans, organization of multidisciplinary case meetings, screening and monitoring of risk factors and symptoms, use of evidence-based guidelines, information support for involved physicians, empowerment of patients through providing education, and enhancing self-management skills [7–9].

There is emerging evidence for the promising effect of case management interventions on reducing hospital (re-)admissions and length of stay in patients with chronic diseases such as asthma [10, 11], diabetes [12, 13], chronic heart failure [13, 14] or established obstructive pulmonary diseases [15, 16]. However, it is unknown whether case management benefits the most common complications in chronic diseases, predominantly depressive symptoms or symptoms of anxiety. In about 40–50% of people with PD, clinically significant depressive

symptoms occur [17]. Symptoms of anxiety is also very common among people with PD, with an average prevalence of 31% [18]. Anxiety and depressive symptoms can contribute to more severe motor and cognitive symptoms, and have a negative impact on the perceived quality of life of people with PD as well as their caregivers [19, 20]. These complications can occur alongside other manifest symptoms or signs of the disease and, in case of PD, might even precede the onset of motor symptoms. Some complications can also occur as a secondary consequence of the original disease itself (for example, a reactive depression) and worsen disease outcomes, creating a vicious cycle. Therefore, this is an important knowledge gap, especially given that people living with a chronic disease such as PD or Alzheimer's disease have identified care coordination as one of their highest priorities. Furthermore, it remains unclear whether the putative beneficial effects of case management on common complications would vary by crucial patient characteristics, such as demographics (i.e., age or gender), disease severity or duration, or critical elements of the intervention (e.g., the number of individual patient contact). Such insights would facilitate a more comprehensive deployment of case management to susceptible subgroups of patients, thereby contributing to a broader shift from "one size fits all"-based healthcare resource allocation to "personalized" medicine.

To close these gaps in knowledge, we conducted a systematic review and meta-analysis to examine: 1) the extent and quality of evidence for the effectiveness of case management interventions on common using insights from various chronic diseases; and 2) to what extent putative effects of case management vary across patient subgroups.

## 2. Methods

This systematic review is guided according to the PRISMA checklist (Preferred Reporting Items for Systematic Reviews and Meta-Analysis) [21].

### 2.1. Search focus

Initially the study was set up to study the effectiveness of case management interventions on common and preventable complications in PD, focusing on the following common potentially preventable complications associated with PD [22, 23]: (1) depressive symptoms and symptoms of anxiety; (2) fractures or injuries caused by falls; (3) swallowing impairment; (4) urinary tract infections; and (5) neuro-psychiatric disorders, including hallucinations. However, an initial search that focused exclusively on case management interventions in people with PD yielded very few results. We, therefore, expanded the scope of this review by including data on case management interventions in people with other chronic health conditions, in whom we hypothesized case management interventions would have similar effects as in people with PD. Although symptoms and prognosis of PD are considerably different from other chronic disease, we believed that there is a substantial overlap in the composition of case management interventions across different chronic disease and that several components are transferrable, such as medication review, a case manager as main contact person and the use of individualized care plans. Specifically, we broadened the search by including all chronic diseases in which case management has been investigated: Alzheimer's disease, asthma, cancer, chronic obstructive pulmonary disease (COPD), chronic heart failure, dementia, diabetes, hypertension, multiple sclerosis, and rheumatoid arthritis.

### 2.2. Search strategy

An initial and limited search for empirical literature was undertaken by one reviewer [ADG] in May 2019 in PubMed to identify important Medical Subject Headings (MeSH) and keywords describing relevant articles. Next, a systematic search for research-based literature was

performed by two independent reviewers [ADG, JMJD] using the identified MeSH terms and keywords. We exclusively focused on published articles referenced in PubMed and Embase online in July 2019. The following MeSH terms were used to identify studies on case management: "Case Management," "Disease Management," "Patient Care Management," "Patient Care Planning," and "Patient-Centered Care," which were combined with relevant keywords. The detailed search strategy for PubMed can be found in Supplemental Data I in S2 File. Finally, the reference lists of included studies were screened to identify studies missed by the search. A verification search was performed in November 2022.

### 2.3. Selection criteria

Abstracts and titles of all obtained studies were independently and systematically examined for the selection criteria by two reviewers [ADG, JMJD], and disagreements were resolved during consensus meetings with a third reviewer [SKLD]. Search limits were applied to include only English articles and those published in a peer-review journal. We included in this systematic review studies that (1) used an observational (prospective and retrospective) or interventional study design; (2) included results on the association of a case management intervention with at least one of the five common and potentially preventable complications; (3) included populations who were diagnosed with one of the selected chronic diseases [aged $\geq$ 18 years]; and (4) described the case management intervention clearly and contained at least three core elements; (5) defined a clear control group, usually receiving usual care; and (6) reported at the minor measures of the distribution of age and gender for the intervention and control group, as potential confounders (Table 1). We excluded studies involving participants living in residential nursing homes, as those participants received 24-hour care and were thus not comparable

**Table 1. Description of predefined core elements of case management interventions derived from previous studies [7, 8].**

| Component | Description |
|---|---|
| Development and review of individualized care plans | Guidance of care through an individual care plan that is tailored to patient's individual needs, preferences and goals. The plan is reviewed and updated on a regular basis. |
| Proactive monitoring of Symptom and/or risk factor | Regular monitoring of risk factors, symptoms and adverse events of therapy implying a proactive follow-up to prevent complications. |
| Medication review and monitoring of therapy adherence | Collecting and documenting all prescribed and self-administered medications a patient uses, including information on patient experience, adverse events, intake experience, and therapy adherence and compliance; accordingly tailoring the medication to individual's medication needs. |
| Multidisciplinary case meetings | Discussing disease management and treatment options of individual patients within scheduled meetings with different other health care providers. |
| Use of evidence-based guidelines | Implementation of evidence-based guidelines valid in daily practice. |
| Delivery of information support for involved health care providers | Educating and training health care professionals on disease management and treatment, including up-to-date knowledge on evidence-based guidelines. |
| Education on disease management | Providing educational materials or trainings to patients on disease management and treatment and/or trainings on enhancing self-management skills. |
| Assessment and providing support on social and financial support | Finding out if the patient and/or the carer is aware of different services for social and financial support; and referral to local community services. |
| Provision of a single point of access | Patients have a central and single point of contact for questions and problems. |

to other populations receiving case management. Also excluded were studies that described only study protocols or conference abstracts.

## 2.4. Data extraction and collection

The quality of the articles was evaluated by two authors (JMJD and ADG) using the Cochrane risk of bias tool for randomized controlled trials (RCTs) [24] and the ROBINS-I tool for non-randomized studies [25]. Two review authors [ADG, JMJD] independently screened and evaluated the studies. The following data were extracted from each study: author, publication year, trial design, country, the aim of the study, study design, distribution of participant characteristics (sex, age, and diagnosis of disease), aspects of case management intervention (Table 1), parts of care received by the control group, follow-up duration, study outcomes, and primary outcome results (mean, confidence intervals, standard deviation, standard error, interquartile range). Standard deviations (SD) were calculated from standard errors (SE). If no SD or SE was reported, authors were contacted and asked to provide additional data. We did not receive further data [23–27] for five articles, so these articles could not be included in the meta-analysis and random-effects regression analysis.

## 2.5. Statistical analysis

We calculated standardized effect sizes for the main results of each study. Subsequently, we conducted meta-analyses using restricted maximum likelihood (REML) and DerSimonian-Laird estimator in a random-effects model, because of significant heterogeneity between estimates across studies as reflected in the I2 statistic. We used funnel plot visualization and Egger's test for funnel plot asymmetry to assess whether there was evidence for small study effects.

In addition, predefined random-effects regression analyses were performed to identify potential effect modifiers if a sufficient number of studies (> 3) was available within a complication category (e.g., depressive symptoms, symptoms of anxiety). We included the following study population characteristics as potential effect modifiers: mean age, percentage of female participants, and disease group (neurodegenerative disease versus other chronic disease). Mean age and percentage of females were treated as numeric variables, whereas disease group was recoded into a binary variable with 0 for "neurodegenerative disease" and 1 for "other chronic disease". We also included the number of case management components (out of nine defined components, which are outlined in Table 1) as a potential effect modifier. Also this variable was recoded into a binary variable with 0 for "was not described as part of the case management intervention" and 1 for "was described as part of the case management intervention". Due to restrictions in the number of potential effect modifiers, we could not include individual components in the random-effects regression analyses for symptoms of anxiety. Instead, we had the number of included case management components as potential effect modifiers. Two studies were excluded from this analysis on depressive symptoms, as the necessary data for regression analysis was not available [26, 27].

Multicollinearity of covariables–the modifying effect of covariables on each other—was assessed using variance inflating factors (VIF). Of note, analyses for symptoms of anxiety and depressive symptoms were conducted by using standardized mean difference as outcome and used the same meta-analysis settings. However, because of statistical constraints due to less studies on symptoms of anxiety which bears the risk to overfit the meta-regression model as the number of studies per examined covariate is low, we had to limit the number of covariables in our meta-regression analysis for symptoms of anxiety. Therefore, we added mean age, sex and number of case management components as covariables for the meta-regression analysis

for symptoms of anxiety. For the meta-regression analysis for depressive symptoms, by contrast, we added the following covariables: mean age, sex, female, duration of follow-up (less than 6 months, 6 to 12 months, and more than 12 months), disease group (neurodegenerative disease versus other chronic disease) and the individual components of case management interventions (development and review of individualized care plans, in-person contact with the case manager, medication review, provision of education on disease management and treatment, self-management support, support and training for healthcare providers, therapy adherence, and use of evidence-based guidelines).

We considered associations with p<0.05 to be statistically significant across meta-analyses and meta-regression analyses. Analyses were conducted in R [28], using packages metafor [29] and ggplot2 for visualization [30].

## 3. Results

### 3.1. Study selection

The combined PubMed and Embase searches yielded 4765 unique records. After abstract review, 57 full-text articles were assessed for eligibility, of which 23 fulfilled our selection criteria. The other 34 were excluded for the following reasons: 25 studies did not provide relevant data on the topic under study; four studies were not identified as case management interventions; two studies dealt with a different health population; two studies did not include original data, and one study presented the same cohort. Two further studies [26, 31] were identified through cross-reference checking and two study [32, 33] was added through verification search, bringing the total to 27 included studies. Fig 1 provides an overview of the search and study selection process. A verification search was performed in November 2022 yielding two additional studies.

### 3.2. Study characteristics

Table 2 describes the characteristics of the included studies. Of the 27 studies included, 23 [26, 27, 31, 33–52] were RCTs and four were non-randomized intervention studies [32, 53–55]. The studies were published between 2002 and 2020 and evaluated case management interventions in various countries. The majority of studies evaluated case management interventions among patients with a single health condition [26, 27, 32, 34–39, 41–44, 46, 47, 49–55], whereas five studies [31, 33, 40, 45, 48] included patients with two or more different chronic diseases, such as heart failure and asthma or COPD. Only one of the included studies was conducted among people with PD. Taken together, the studies included 3752 participants ascribed to case management interventions. Mean age ranged from 57 years to 80 years, with an average age across all studies of 65 years, with 51% being female among the 25 studies [26, 31–55] reporting gender. Three thousand six hundred eighty-two participants were ascribed to the usual care group. The mean age ranged from 51 years to 78 years, with an average age across all studies of 51 years, with 51% being women.

### 3.3. Risk of bias and quality assessment

Of the four non-randomized intervention studies, three were rated as carrying a high risk of bias and one as moderate. Of the RCTs, only eight studies [26, 31, 33, 34, 36, 37, 40, 51] were judged to be of excellent methodological quality and at low risk of carrying bias. The remaining 15 studies were of low or moderate quality with a high or unknown risk of bias among several domains. However, most RCTs were rated as having severe or unfamiliar trouble on the parts of blinding participants and blinding of outcome assessment, which is less applicable to

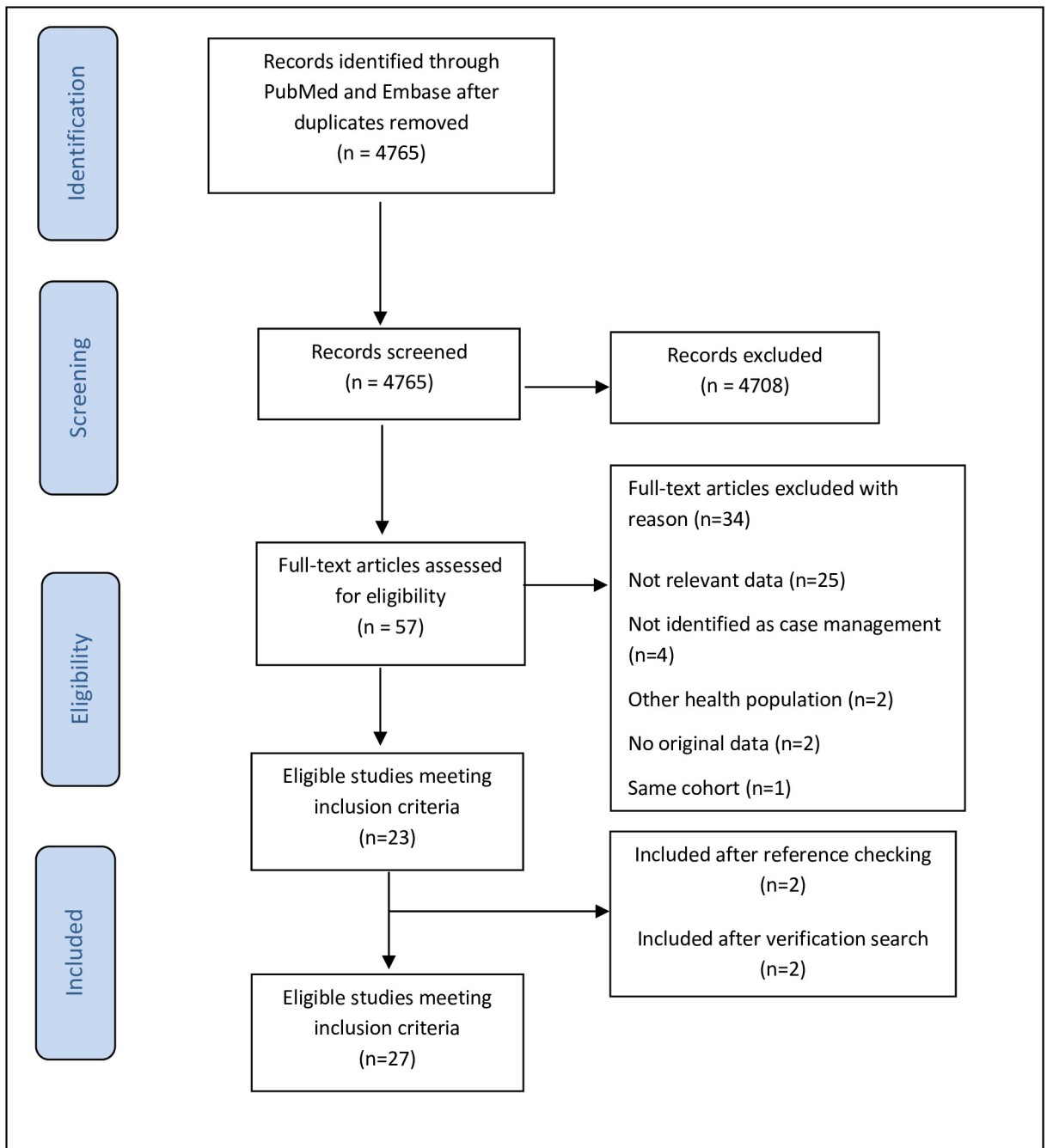

**Fig 1. Study flow diagram.**

this kind of intervention. Details on the quality assessment are presented in Supplemental Data II in S2 File.

### 3.4. Components of case management intervention

Across the 27 included studies, there was substantial heterogeneity across the content and duration of the various case management interventions. Table 3 displays the different

**Table 2. Main study characteristics.**

| Outcome | Study (Author, year) | Country | Study design | Chronic disease of participants | N (% women) | Mean age in years (SD) | Outcome measures | | Summary of results | Follow-up in months | Quality score*** |
|---|---|---|---|---|---|---|---|---|---|---|---|
| | | | | | | | Feelings of anxiety | Depressive symptoms | | | |
| Anxiety Depression | Stoop et al. 2015 [48] | The Netherlands | RCT | Heart Failure and/or COPD and/or asthma | I: 23 (57) C: 23 (47) | I: 66.2 (11.7) C: 54.8 (13.9) | GAD-7 | PHQ-9 | The intervention group reported lower symptoms of depression, but no significant differences was found; the intervention group reported less anxiety scores with a significant difference | 18 | 4/7 |
| Anxiety Depression | Egan et al. 2002 [38] | Australia | RCT | COPD | I: 33 (64) C: 33 (40) | I: 67.2 (x) C: 67.8 (x) | HADS | HADS | The intervention group experienced an improvement in the level of anxiety, but no significant difference was detected between groups; the intervention had no significant impact on depression | 1; 3 | 4/7 |
| Anxiety Depression | Hernandez et al. 2015 [41] | Spain | RCT | COPD | I: 71 (17) C: 84 (14) | I: 73 (-) C:75 (-) | HADS | HADS | The intervention group showed less depression scores with a significant difference; and less anxiety symptoms with no significant difference | 12 | 5/7 |
| Anxiety Depression | Mertz et al. 2017 [44] | Denmark | RCT | Breast cancer | I: 25 (100) C: 25 (100) | I: 50.9 (12.1) C: 55.0 (9.8) | HADS | HADS | The intervention group experienced an improvement in depressive and anxiety scores with a significant difference | 6; 12 | 5/7 |
| Anxiety Depression | Rose et al. 2017 [47] | NR | RCT | COPD | I: 236 (50) C: 234 (56) | I: 71 (9.2) C: 71 (9.7) | HADS | HADS | No significant group differences were found in depression scores | 3; 6; 12 | 4/7 |
| Anxiety Depression | Titova et al. 2017 [49] | Norway | RCT | COPD | I: 91 (57) C: 80 (58) | I: 73.6 (9.2) C: 72.2 (9.4) | HADS | HADS | Intervention group experienced decrease in anxiety and depression scores with a significant difference | 24 | 2/7 |
| Anxiety Depression | Tsuchihashi-Makaya et al. 2013 [50] | Japan | RCT | Heart Failure | I: 79 (47) C: 82 (40) | I: 76.9 (10.9) C: 75.8 (12.1) | HADS | HADS | Intervention group experienced decrease in anxiety and depression scores with a significant difference | 2; 6; 12 | 2/7 |
| Anxiety | Avci et al. 2019 [53] | Turkey | Non-randomized | Colorectal cancer | I: 60 (26) C: 60 (32) | I: 59.0 (11.5) C: 61.6 (12.3) | STAI | - | The intervention group experienced less frequently anxiety than the control group with a significant difference. | ** | 3/9 |
| Depression | Gabbay et al. 2013 [39] | USA | RCT | Diabetes* | I: 232 (62) C: 313 (55) | I: 58 (11.41) C: 58 (11.34) | - | CES-D | The intervention led to improved depression scores | 24 | 3/7 |

(Continued)

**Table 2.** (Continued)

| Outcome | Study (Author, year) | Country | Study design | Chronic disease of participants | N (% women) | Mean age in years (SD) | Outcome measures | | Summary of results | Follow-up in months | Quality score*** |
|---|---|---|---|---|---|---|---|---|---|---|---|
| | | | | | | | Feelings of anxiety | Depressive symptoms | | | |
| Depression | Miklavcic et al. [2020] [33] | Canada | RCT | Diabetes* | I: 70 (62.9) | I: - | - | CESD-10 | No significant group differences were found in depression scores | 6 | 6/7 |
| | | | | | C: 28 (45.2) | C: - | | | | | |
| Depression | Steel et al. 2016 [27] | USA | RCT | Cancer | I: 144 (-) | I + C: 61 (-) | - | CES-D | Depression scores decreased in the intervention group | 6 | 5/7 |
| | | | | | C: 117 (-) | | | | | | |
| Depression | Callahan et al. 2006 [35] | USA | RCT | Alzheimer's Disease | I: 39 (46) | I: 77.4 (5.9) | - | Cornell scale | The intervention had no significant impact on depression scores | 6; 12; 18 | 6/7 |
| | | | | | C: 27 (39) | C: 77.7 (5.7) | | | | | |
| Depression | Kroenke et al. 2010 [43] | USA | RCT | Cancer | I: 202 (63) | I: 58.7 (11.0) | - | HSCL-20 | The intervention group experienced an improvement in depressive symptoms with a significant difference | 1; 3; 6; 12 | 5/7 |
| | | | | | C: 203 (72) | C: 59.0 (10.6) | | | | | |
| Depression | Bogner et al. 2012 [34] | USA | RCT | Diabetes* | I: 94 (70) | I: 57.8 (9.4) | - | PHQ-9 | The intervention group was more likely to achieve remission of depression compared to the control group with a significant difference. | 4 | 7/7 |
| | | | | | C: 88 (66) | C: 57.1 (9.6) | | | | | |
| Depression | Chen et al. 2018 [36] | China | RCT | Heart Failure | I: 31 (29) | I: 61.1 (14.2) | - | PHQ-9 | The intervention group experienced an improvement in depressive symptoms with a significant difference | 3; 6 | 7/7 |
| | | | | | C: 31 (52) | C: 62.4 (14.9) | | | | | |
| Depression | Connor et al. 2019 [37] | USA | RCT | Parkinson's disease | I: 162 (4) | I: 69.6 (10.1) | - | PHQ-9 | The intervention group experiences less depression feeling, but no significant difference was found | 6; 12; 18 | 7/7 |
| | | | | | C: 166 (1) | C: 71.3 (9.2) | | | | | |
| Depression | Crowley et al. 2016 [54] | United Kingdom | Non-randomized | Diabetes* | I: 23 (0) | I: 60 (8.4) | - | PHQ-9 | The intervention had no impact on depression scores | 3; 6 | 6/9 |
| | | | | | C: 23 (8) | C: 60 (9.2) | | | | | |
| Depression | Ell et al. 2008 [26] | USA | RCT | Cancer | I: 242 (84) | I: - | | PHQ-9 | The intervention group experienced an improvement in depressive symptoms and the difference was borderline significant | 6; 12 | 6/7 |
| | | | | | C:230 (86) | C: - | | | | | |
| Depression | Gellis et al. 2012 [40] | USA | RCT | Heart Failure and/or COPD | I: 57 (63) | I: 80.1 (7.8) | - | PHQ-9 | The intervention group experiences reductions in depressive symptoms with a significant difference | 12 | 7/7 |
| | | | | | C: 58 (69) | C: 78.3 (6.9) | | CES-D | | | |

(Continued)

**Table 2.** (Continued)

| Outcome | Study (Author, year) | Country | Study design | Chronic disease of participants | N (% women) | Mean age in years (SD) | Feelings of anxiety | Depressive symptoms | Summary of results | Follow-up in months | Quality score*** |
|---|---|---|---|---|---|---|---|---|---|---|---|
| | | | | | | | Outcome measures | | | | |
| Depression | Johnson et al. 2014 [55] | Canada | Non-randomized | Diabetes* | I: 95 (61) | I: 57.0 (10.5) | - | PHQ-9 | The intervention group experienced greater improvements in depression scores with a significant difference | 6; 12 | 7/9 |
| | | | | | C: 71 (56) | C: 63.4 (11.3) | | | | | |
| Depression | Kalter-Leibovici et al. 2017 [42] | Israel | RCT | Heart Failure | I: 682 (31) | I: 70.8 (11.6) | - | PHQ-9 | The intervention group was less likely to experience moderate-to-severe depression symptoms | 12 | 5/7 |
| | | | | | C: 678 (24) | C: 70.7 (11.0) | | | | | |
| Depression | Morgan et al. 2013 [45] | Australia | RCT | Diabetes* and/or | I: 170 (48) | I: 68.0 (11.7) | - | PHQ-9 | Depression scores were significantly lower for patients in the intervention group compared to the control group | 12 | 4/7 |
| | | | | Heart Failure | C: 147 (45) | C: 67.6 (11.2) | | | | | |
| Depression | Riegel et al. 2006 [46] | USA | RCT | Heart Failure | I: 69 (58) | I: 71.6 (10.8) | - | PHQ-9 | No significant group differences were found in depression scores | 6 | 6/7 |
| | | | | | C: 65 (49) | C: 72.7 (11.2) | | | | | |
| Depression | Wu et al. 2018 [52][1] | USA | RCT | Diabetes* | I: 461 (59) | I: 51.9 (9.3) | - | PHQ-9 | The intervention group experienced an improvement in depressive symptoms with a significant difference | 6, 12 | 3/7 |
| | | | | | C: 416 (70) | C: 55.2 (9.2) | | SCL-20 | | | |
| Depression | Katon et al. 2010 [31] | USA | RCT | Diabetes* and/or | I: 106 (48) | I: 57.4 (10.5) | - | SCL-20 | The intervention group experienced an improvement in depressive symptoms with a significant difference | 6,12 | 6/7 |
| | | | | Heart Failure | C: 108 (56) | C: 56.3 (12.1) | | | | | |
| Depression | Williams et al. 2004 [51] | USA | RCT | Diabetes* | I: 205 (54) | I: 70.1 (6.9) | - | SCL-20 | The intervention group experienced an improvement in depressive symptoms with a significant difference | 3; 6; 12 | 7/7 |
| | | | | | C: 212 (53) | C: 70.3 (7.1) | | | | | |
| Depression | Lu et al. 2020 [32] | Taiwan | Non-randomized | Rheumatoid Arthritis | I: 50 (76) | I: 56.6 (10.3) | - | TDQ | Depression scores were significantly lower for patients in the intervention group compared to the control group | 3; 6 | 3/9 |
| | | | | | C: 46 (89) | C: 50.7 (10.8) | | | | | |

NR: not reported

*:Diabetes mellitus type 2; N: number of participants; RCT: Randomized Controlled Trial; I: Intervention group; -: not measured; C: Control group; CES-D: Center for Epidemiological Studies; GAD-7: 7-item General Anxiety Disorder; HADS: Hospital Anxiety/Depression Scale; PHQ-9: 9-item Patient Health Questionnaire; SCL-20: Symptom Checklist-20; TDQ: Taiwanese Depression Questionnaire

** no clear follow-up time frame

*** quality score on basis of Cochrane Handbook for Systematic Reviews of Interventions and ROBINS-I tool.

[1] next to a case management intervention and a usual care group, this study included a third group receiving technology facilitated care; as this exceeds the scope of this review, this group was not included in further analyses.

**Table 3. Specific case management strategies used per study interventions.**

| Study (Author, year) | Development & reviewing of individualized care plans | Screening and monitoring | | | MCC* | Use of evidence-based guidelines | Support/ training for involved HCPs | Education | | Assistance social/ financial support | Continuous contact | | |
|---|---|---|---|---|---|---|---|---|---|---|---|---|---|
| | | Therapy adherence | Medication review | Monitoring | | | | Education/ Information material | Self-management skills | | Tele-phone | Mail/ chat | In-person |
| Avci et al. 2019 [53] | NR | NR | NR | + | NR | NR | NR | + | + | NR | + | + | + |
| Bogner et al. 2012 [34] | NR | + | + | + | NR | + | + | + | NR | NR | + | NR | + |
| Callahan et al. 2006 [35] | + | + | + | + | + | + | + | + | + | + | + | NR | + |
| Chen et al. 2018 [36] | NR | + | NR | + | NR | NR | NR | + | + | NR | + | NR | + |
| Connor et al. 2019 [37] | + | NR | NR | + | NR | + | + | NR | + | NR | + | NR | NR |
| Crowley et al. 2016 [54] | NR | + | + | + | NR | NR | NR | + | + | NR | + | NR | NR |
| Egan et al. 2002 [38] | + | NR | + | + | + | NR | NR | + | + | + | + | NR | NR |
| Ell et al. 2008 [26] | + | NR | NR | + | NR | + | NR | + | + | + | + | NR | + |
| Gabbay et al. 2013 [39] | NR | + | + | + | NR | NR | + | + | NR | NR | + | + | + |
| Gellis et al. 2012 [40] | NR | + | NR | + | NR | + | + | + | + | NR | + | NR | + |
| Hernandez et al. 2015 [41] | + | NR | NR | + | NR | NR | + | + | + | NR | + | NR | + |
| Johnson et al. 2014 [55] | + | + | NR | + | + | + | + | NR | + | NR | + | NR | + |
| Kalter-Leibovici et al. 2017 [42] | | + | + | + | NR | NR | NR | NR | + | NR | + | NR | + |
| Katon et al. 2010 [31] | NR | + | + | + | NR | + | NR | + | + | NR | NR | NR | + |
| Kroenke et al. 2010 [43] | NR | + | + | + | NR | + | NR | + | NR | NR | + | NR | NR |
| Lu et al. 2020 [32] | + | + | + | + | NR | + | + | + | + | NR | + | NR | NR |
| Mertz et al. 2017 [44] | NR | NR | NR | + | NR | NR | NR | NR | + | NR | + | NR | + |
| Miklavcic et al. 2020 [33] | NR | NR | NR | NR | + | + | + | + | + | + | NR | NR | + |
| Morgan et al. 2013 [45] | + | + | NR | + | NR | + | + | + | + | NR | NR | NR | + |
| Riegel et al. 2006 [46] | NR | + | NR | + | NR | + | NR | + | + | NR | + | NR | NR |

(*Continued*)

**Table 3.** (Continued)

| Study | Development & reviewing of individualized care plans | Screening and monitoring | | | MCC* | Use of evidence-based guidelines | Support/ training for involved HCPs | Education | | Assistance social/ financial support | Continuous contact | | |
|---|---|---|---|---|---|---|---|---|---|---|---|---|---|
| (Author, year) | | Therapy adherence | Medication review | Monitoring | | | | Education/ Information material | Self-management skills | | Tele-phone | Mail/ chat | In-person |
| Rose et al. 2017 [47] | + | + | NR | + | NR | NR | + | + | + | NR | + | NR | NR |
| Steel et al. 2016 [27] | NR | + | + | + | NR | NR | + | + | + | NR | + | NR | + |
| Stoop et al. 2015 [48] | NR | NR | NR | + | NR | + | NR | + | + | NR | NR | NR | + |
| Titova et al. 2017 [49] | + | NR | NR | NR | NR | NR | NR | + | + | NR | + | NR | + |
| Tsuchihashi-Makaya et al. 2013 [50] | NR | + | NR | + | NR | + | NR | + | NR | NR | + | NR | + |
| Williams et al. 2004 [51] | + | + | + | + | + | + | + | + | NR | NR | + | NR | + |
| Wu et al. 2018 [52] | NR | + | NR | + | NR | + | + | + | + | NR | + | NR | + |

NR: Not reported or has not been part of the case management intervention;

*MCC: Multidisciplinary Case Conference

strategies used in each study. Typical components of case management among the 27 studies were (1) regular telephone contacts combined with in-person visits; (2) monitoring of signs, symptoms, and risk factors; (3) ensuring therapy adherence; and (4) providing educational support on disease management and treatment or training on self-management skills. However, the content and structure of these components varied highly among these studies. For instance, in-person home visits ranged from developing and discussing a therapeutic plan with the patient in one study [41] to monitoring changes in signs and symptoms and reviewing patients' safety in their home environment in another study [36]. The location of in-person visits also varied from the patient's home to a clinical setting. Most studies were conducted through a combination of in-person and telephone contacts, with only seven studies [32, 37, 38, 43, 46, 47, 54] reporting an intervention exclusively conducted through telephone contact.

More commonly reported case management interventions, assisted with social and financial support, organized multidisciplinary case meetings and medication reviews, and developed individualized care plans. In only four studies, case management interventions also included the support of informal carers [33, 35, 38, 42]. And of these, only one study had a regular assessment of the carer's physical health and education on the carer's coping skills and educating carers in the disease management [35].

Furthermore, several studies incorporated technological support systems, which supported the implementation of case management strategies and offered new possibilities. For instance, a web-based service facilitates communication between the patient and the care team, schedules patient contacts, and keeps track of progress and current disease treatment [35]. In a different study, a web-based collaborative intervention facilitated peer-to-peer support for patients with cancer through a chat room connecting all enrolled participants [27].

Of the 27 studies, 23 [27, 31–33, 35–46, 49–55] reported that case management interventions were delivered by a nurse case manager or a team consisting of a nurse case manager and other health care specialists. In one study, [26], the case manager was a depression clinical specialist without further clarification of the specialist's background. In another one, [34], two research coordinators fulfilled the role of care managers. In the other two studies, the background of the case manager was not specified any further by [47, 48]. The length of follow-up ranged from one and a half months to 24 months, with 18 studies reporting a 12-month or even longer follow-up period (Table 2).

## 3.5. Overview of outcome measurements

Results of the narrative data synthesis are summarized and presented in Table 2. Nearly all included studies evaluated the effectiveness of case management interventions on depressive symptoms, whereas symptoms of anxiety was addressed in only eight studies [38, 41, 44, 47–50, 53]. None of the included studies reported falls, urinary tract infections, swallowing impairment, or hallucinations. Of note, we included individuals with depressive and anxiety symptoms, not necessarily reflecting individuals with a DSM-5 diagnosis such as major depressive disorder (MDD).

**3.5.1. Effect of case management on symptoms of anxiety.** Eight studies (n = 1239 participants) reported outcomes on symptoms of anxiety [38, 41, 44, 47–50, 53]. The most commonly used scale was the Hospital Anxiety and Depression Scale (HADS) [38, 41, 44, 47, 49, 50], followed by 7-item Generalized Anxiety Disorder [48], and State Trait Anxiety Index (STAI) [53]. Six studies reported sufficient data and were included in a random-effects meta-analysis, the results of which revealed a significant effect of case management interventions in decreasing symptoms of anxiety (Standardized Mean Difference [SMD] = - 0.47; 95% confidence interval [CI]: -0.69, -0.324 with moderate heterogeneity ($I^2$ = 51.9%) (Fig 2).

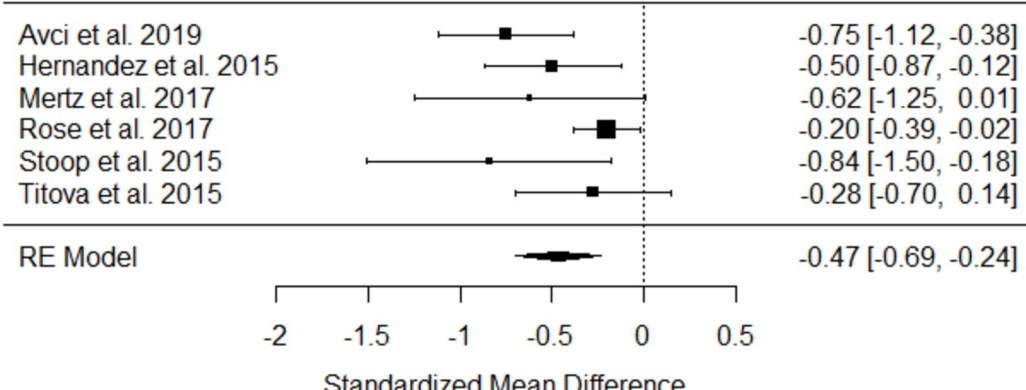

**Fig 2. Forest plot on the effect of case management interventions on feelings of anxiety (random-effects model).**

Variance inflating factor (VIF) analysis showed no evidence of multicollinearity (all VIFs <5). A tendency for lower SMD was found for studies with a higher percentage of females [standardized regression coefficient β = 0.01, p value = 0.05]. No significant effect was found for mean age of intervention group [β = 0.02, p = 0.13] and number of components were [β = 0.13, p = 0.21].

**3.5.2. Effect of case management on depressive symptoms.** With the exception of one study [53], all studies (n = 7314 participants) reported depressive symptoms measurements using a total of six depression scales, including the 9-item Patient Health Questionnaire (PHQ-9) [26, 34, 36, 37, 40, 42, 45, 46, 48, 52, 54, 55], the Hospital Anxiety/Depression Scale (HADS) [38, 41, 44, 47, 49, 50], the Center for Epidemiologic Studies Depression scale (CES-D) [27, 33, 39, 40], 20-item Symptom Checklist (SCL-20) [31, 43, 50, 52], the Cornell Scale for depression [35] and Taiwanese Depression Questionnaire [32].

Twenty studies reported sufficient data and were included in the meta-analysis (Fig 3). Two studies [40, 52] reported two different depression outcomes (CES-D/SCL-20 PHQ-9), but for analytical reasons, only one (PHQ-9) was included in the meta-analysis. A random-effects meta-analyses revealed a significant effect of case management intervention on depressive symptoms (SMD = - 0.48; CI: -0.71, -0.25), but heterogeneity was high ($I^2$ = 92.3%). A funnel plot and Egger's test of funnel plot asymmetry (p = 0.57) showed no evidence for publication bias.

Variance inflating factor (VIF) analysis showed no evidence of multicollinearity (all VIFs<5). There were no significant population or disease characteristics: neurodegenerative diseases (reference: other chronic disease) [β = 0.20, p = 0.80], mean age of intervention group [β = -0.01, p = 0.76] and percentage of females assigned to case management intervention [β = -0.01, p = 0.43]. Moreover, no significant effect was found for any of the individual case management components: development and review of individualized care plans [β = -0.60, p = 0.21]; in-person contact with the case manager [β = -0.09, p = 0.89]; medication review [β = 0.39, p = 0.47]; provision of education on disease management and treatment [β = -0.08, p = 0.91]; self-management support [β = -0.24, p = 0.70]; support and training for healthcare providers [β = 0.05, p = 0.92]; therapy adherence [β = 0.73, p = 0.92] and use of evidence-based guidelines [β = 0.27, p = 0.52].

## 4. Discussion

This systematic review and meta-analysis show that case management is more effective than usual care at reducing depressive symptoms and symptoms of anxiety, two common

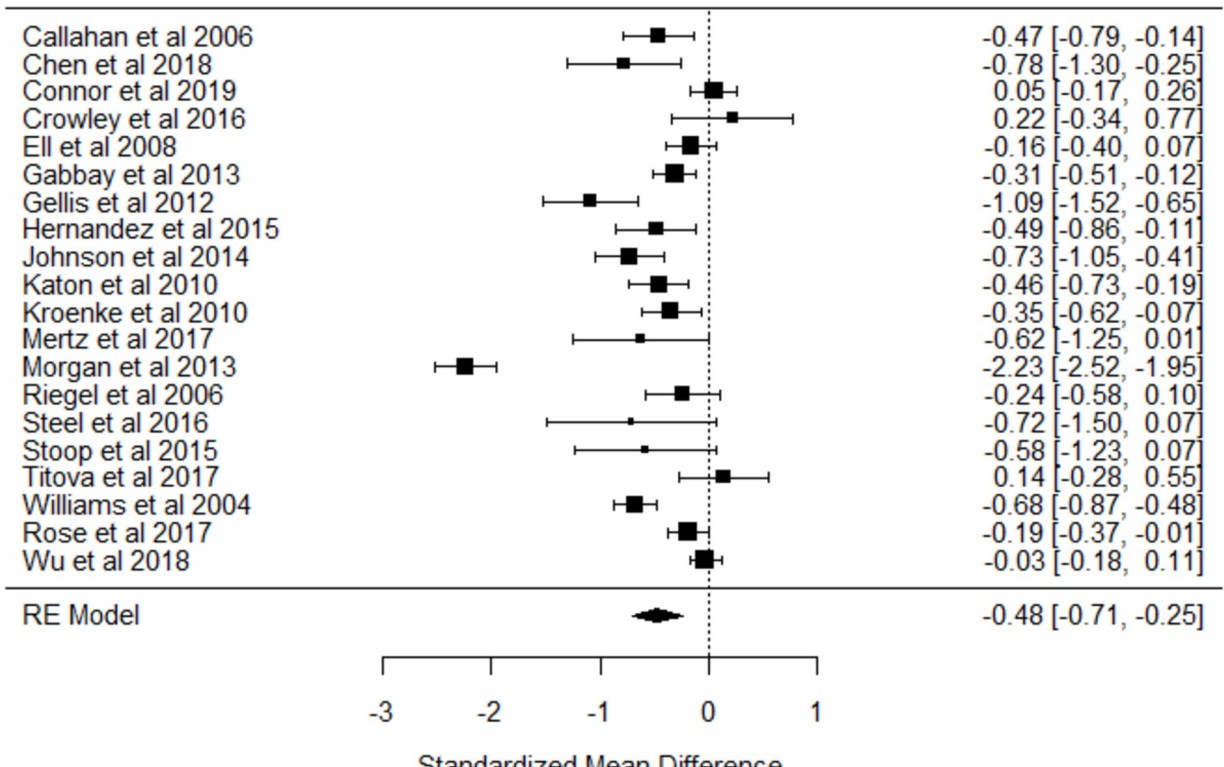

**Fig 3. Forest plot on the effect of case management interventions on depressive symptoms (random-effects model).**

preventable complications associated with many chronic diseases, including PD. This effect persisted for less complex case management interventions (whereby complexity was based on the number of included elements), across countries, for different chronic diseases and given the nearly equal representation of female and male participants across both genders. Contrary to our hypotheses, we found no evidence for effect modification by case management intervention, study population characteristics, or duration of follow-up. Our data showed that the beneficial effects of case management interventions on symptoms of anxiety were somewhat more distinct in studies with a higher percentage of men and studies with a relatively old population. However, both observations were not statistically significant. No similar trends were observed for the effects of case management interventions on depressive symptoms. We did not identify any studies reporting on falls, hallucinations, swallowing impairment, and urinary tract infections, so the effect of case management on those complications remains unclear. Although initially we specifically looked at common complications in PD, only one of the included interventions was conducted in people with PD [37], indicating that case management interventions in PD are still rare. Although we specifically looked at common complications in PD in the first place, only one of the included interventions was actually conducted in people with PD [37], indicating that research on the effectiveness of case management interventions in PD is still rare. We want to emphasize that to our knowledge case management interventions are fairly common in clinical practice, but research with well-designed RCT's or even cohort studies on the topic of case management are rare. The randomized study reported that a PD nurse-led care management intervention among veterans led to better adherence to quality-of-care indicators. The screening instrument showed significant improvement among the intervention group compared to usual care [37]. In this trial, the PD nurse used four strategies to

specify the PD problems of each veteran and to develop an action plan: (1) a telephone-administered assessment to identify 28 problem areas; (2) evidence-based care protocols and, if not available, use of expert consensus on care management; (3) patient portal for communication purposes; and (4) documentation templates to provide care that is patient-centered and coordinated. However, this study is limited by geographical and time factors.

Our study extends the findings of previous studies indicating a beneficial effect of case management on patients' clinical health outcomes and functioning in everyday life [5, 56, 57]. While previous research on case management has focused on its effects on reducing hospital (re-)admissions, length of stay, and costs, little research has been done regarding its potential to reduce disease complications. Our systematic review addresses this gap in knowledge in a disease in which case management will likely have high potential. The findings of this systematic review favor case management interventions over usual care and suggest that even less complex case management interventions have a beneficial effect on symptoms of anxiety and depression, which have an immense impact on the quality of life. Our meta-analysis suggests a modest effect size of case management interventions in reducing feelings symptoms of anxiety and depressive symptoms. However, this finding needs to be interpreted with caution as the magnitude of reduction in symptoms of anxiety and depression scores does not automatically translate into a lower risk of anxiety and depressive disorder. Large-scale studies assessing the personal impact of case management on clinically relevant measures of anxiety and depressive symptoms are therefore warranted. Moreover, several indirect working mechanisms may have contributed to the beneficial effect on depressive symptoms and stress (Fig 4), which can immensely impact on the quality of life and even mortality in patients with PD [58–60]. First, patients across studies received personalized management of their chronic health conditions, including individually tailored health information and problem-solving strategies provided by a case manager. Second, the availability of one main contact person for new issues and establishing a personal relationship between patients and case managers may have helped reduce symptoms of anxiety and depressive symptoms. Notably, previous research on improving PD care revealed that having a single point of access was rated as the top priority by people with PD [2]. Third, having little information about the rate of disease progression and treatment

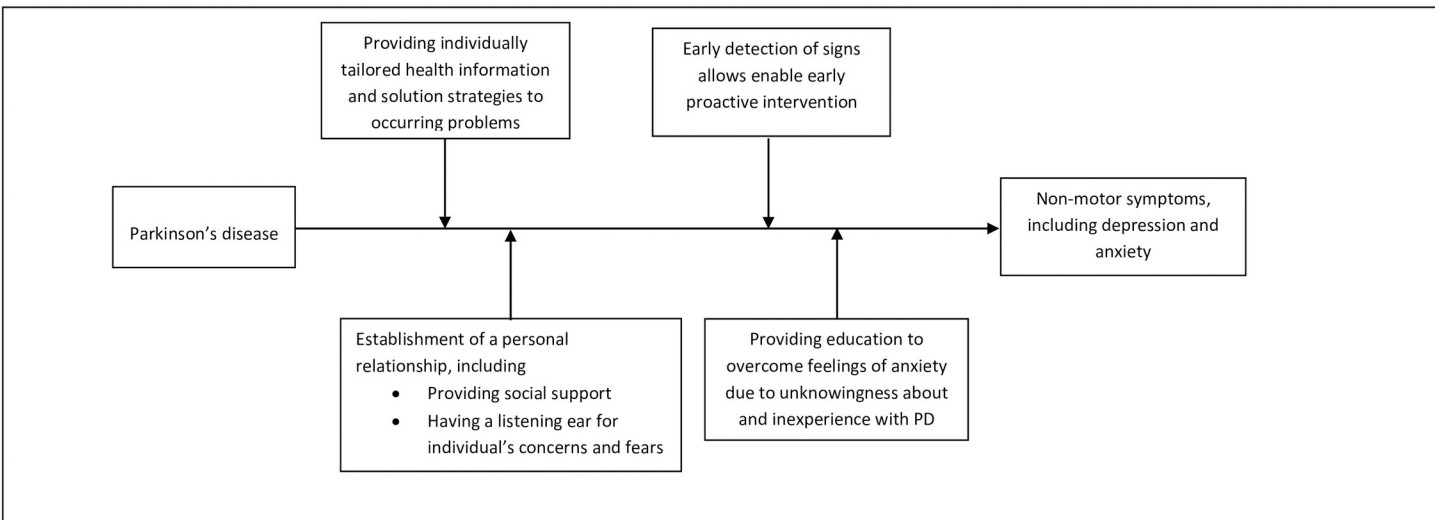

**Fig 4. Mediators of the effect of case management on reduction of depressive symptoms and feelings of anxiety: Hypothetical framework informed by this meta-analysis.**

options is known to enhance symptoms of anxiety in patients. Patient education eventually allows for more shared decision-making and, thereby, a treatment better tailored to a patient's individual needs and coping behavior, which, in turn, might alleviate symptoms of anxiety.

Several methodological considerations need to be considered. First, heterogeneity across studies was high, and effect estimates varied substantially between studies, which might have affected the results of our meta-regression, leading to the inability to identify any covariables of our effect estimates. Furthermore, since only two studies [39, 49] followed patients for longer than two years, the long-term effectiveness of case management interventions still needs to be determined. Second, only a few included studies were of good methodological quality. In particular, limited blinding of outcome assessment, insufficient details on the specification of the efficacy of specific case management elements, and the lack of participant selection limited our ability to assess case management impact more accurately. In addition, as a systematic search strategy cannot screen full-text articles, the risk remains that the used search strategy needs to capture relevant articles. Eggers et al. [61], for instance, conducted an RCT with a nurse case-manager-led intervention with patients with PD and reported on the effectiveness of CM in reducing hospitalization caused by falls. As these findings have yet to be reported in the study's abstract, this study was not captured through our search strategy. Furthermore, the use of antidepressant or anti-anxiety medication may have affected the results. Still, since some of the original studies included in this meta-analysis did not report sufficient information on medication use, we could not assess this in the meta-regression analysis. For example, an intensive and proactive medication review schedule as part of a case management intervention might have contributed to better outcomes in the intervention group. Finally, although case management interventions as part of research trials are generally free of cost and well-accessible for recruitment purposes, the merits of case management interventions in daily life might be influenced by cost constraints and limited access to care. Specifically, this would hamper structured clinical take-up in private healthcare or increase the social disparity.

Our data showed that the beneficial effects of case management interventions on symptoms of anxiety were somewhat more distinct in studies with a higher percentage of men and studies with a relatively old population. However, both observations were not statistically significant. No similar trends were observed for the effects of case management interventions on depressive symptoms. This systematic review identified the need for further research into most compelling case management interventions and components, the optimal intensity and frequency of the individual case management strategies, and their interaction with patient characteristics. Once this knowledge becomes available, case management interventions can be implemented that are better tailored to individual needs and, as such, presumably more effective. Moreover, this systematic review revealed that case management implementation is more common among certain chronic diseases than others; the commonest ones were diabetes and heart failure, while only one included study concerned patients with PD. It should be noted that anxiety and depression in PD can be both comorbid and arise due to disease burden. It is unknown whether these different causes should be treated differently and whether the efficacy of case management interventions differs for various reasons or disease groups. As we do not have sufficient data to evaluate this, future case management trials in PD are warranted. In addition, future research is warranted to improve the current evidence base for case management effects on symptoms of anxiety and depressive symptoms in persons with a chronic disease, specifically in persons with PD.

## Supporting information

**S1 Checklist. PRISMA 2009 checklist.**
(PDF)

**S1 File.**
(XLSX)

**S2 File.**
(PDF)

## Author Contributions

**Conceptualization:** Angelika D. Geerlings, Sirwan K. L. Darweesh.

**Methodology:** Angelika D. Geerlings, Jules M. Janssen Daalen, Sirwan K. L. Darweesh.

**Supervision:** Bastiaan R. Bloem.

**Visualization:** Angelika D. Geerlings.

**Writing – original draft:** Angelika D. Geerlings, Jules M. Janssen Daalen, Jan H. L. Ypinga.

**Writing – review & editing:** Jan H. L. Ypinga, Bastiaan R. Bloem, Marjan J. Meinders, Marten Munneke, Sirwan K. L. Darweesh.

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
