## [Decision Letter · Decision Letter 0]

10 Oct 2022

PONE-D-22-05522Effectiveness of case management interventions in reducing common and potentially preventable complications associated with Parkinson’s disease:A systematic review and meta-analysisPLOS ONE

Dear Dr. Darweesh,

Thank you for submitting your manuscript to PLOS ONE. After careful consideration, we feel that it has merit but does not fully meet PLOS ONE’s publication criteria as it currently stands. Therefore, we invite you to submit a revised version of the manuscript that addresses the points raised during the review process.

We look forward to receiving your revised manuscript.

Kind regards,

Vincenzo De Luca

Academic Editor

PLOS ONE

Journal Requirements:

3. Please ensure that you refer to Figure 4 in your text as, if accepted, production will need this reference to link the reader to the figure.

Reviewers' comments:

Reviewer's Responses to Questions

**Comments to the Author**

1. Is the manuscript technically sound, and do the data support the conclusions?

Reviewer #1: Yes

Reviewer #2: Partly

Reviewer #3: Yes

2. Has the statistical analysis been performed appropriately and rigorously? 

Reviewer #1: Yes

Reviewer #2: I Don't Know

Reviewer #3: Yes

3. Have the authors made all data underlying the findings in their manuscript fully available?

Reviewer #1: No

Reviewer #2: No

Reviewer #3: Yes

4. Is the manuscript presented in an intelligible fashion and written in standard English?

Reviewer #1: No

Reviewer #2: Yes

Reviewer #3: Yes

5. Review Comments to the Author

Reviewer #1: Need to embed the tables that are reference. Clarify which statistical tests were used. Provide the figure of the PRIMSA with. Changes were recommended to adjust the grammatical errors based upon the English (US) vernacular.

Reviewer #2: From the perspective of investigating how case management might benefit PD — the study makes sense. However, due to the lack of available empirical studies, the content of the study has been shifted to focus on what is available in the literature about chronic conditions. Thus it might be relevant to add more background on the current knowledge of how PD was related to depression and anxiety, e.g., prevalence. — are they comorbid or one leading to another? Also if the study introduction may need to address some background on chronic disease and how they relate to depression and anxiety.

Although the study was motivated by reviewing case management associated with PD — it in fact reviewed studies on case management involving multiple chronic conditions (all claimed to be associated with PD). Perhaps the title and abstract should better reflect the scope – so that readers not interested in PD but interested in chronic conditions and outcomes such as depression and anxiety would still be finding this paper for reference. The major contribution seems to be reviewing case management impacts on depression and anxiety for patients with other chronic diseases.

It is not entierely claar what varaibles were direclty invovled in the analysis. It might be better to include the input for the analysis as supplementary so the results can be verified easily.

A note about depression measures using different scales. It is unclear whether improvements measured on different scales can be compared as percentage improvements. E.g., the raw scores are ordinal but not depicting a linear increase of depression symptoms on depression scales. The depression scales are usually validated against some cutoff values to classify patients as low, moderate, or high risks. A reduction of anxiety from high risk to moderate risk might be more meaningful than the reduction of anxiety by 2 points (or x%), as the patient’s severity of symptoms may not meaningfully change.

Another concern is that follow-up periods vary — this is another variable that would be difficult to interpret in conjunction with the depression measurement. E.g., score change at 3 months may not be that much different than score change at 6 months or 12 months. This variable would need to be treated as cetegorical variable.

An interesting note is that some of the chronic conditions included are quite severe and may lead to more mental disorder symptoms and patients might be taking medications for depression and anxiety symptoms.

The heterogeneity of the studies included is also somewhat concerning — as patients’ experiences from the different medical systems may vary drastically and is beyond the controlled factors (e.g., Does the health care system covers the patient’s bill? Can the patient access private health care, or everyone has to wait their turn? ).

Another note is about clarifying whether depression and anxiety — were clinically validated comorbid conditions (e.g., major depressive disorder, generalized anxiety disorder) or were viewed as an indicator of symptoms only. If the data is available in raw scores — it would be important to inspect whether the improvements were exhibited in patients with comorbid clinical conditions or patients with symptoms yet not met the clinical threshold for depression and anxiety. If the majority of patients have clinical depression or anxiety then it would be important to examine whether the improvement carries clinical significance (e.g., remission achieved? )

Taken together, the study seems to offer good information on what has been done in validating case management interventions on mental health outcomes such as depression and anxiety, but the relation of the study content to PD is tangential. The study may offer limited insight on the prevention of complications – e.g., reduced symptoms or harm reduction may not be the same as prevention (unless the concept of complication is clearly defined), and complications might be caused by comorbidity where the temporal sequence is difficult to establish (e.g., someone developed depression after heart failture, versus someone had major depressive disorder already then later had a heart failure.). There would be many opportunity to further improve the manuscript and the manuscript might benefit from having reviewers with expertise in the domain of chronic diseases and it’s associated treatments for more domain-specific feedback on the discussion.

Reviewer #3: The manuscript by van Halteren and colleagues used a systematic review approach, according to the PRISMA, to investigate the effectiveness of case management interventions on patients with chronic health conditions. The study explored case management interventions across studies identified throughout a review process. The paper is well written, and the methods are clear and straight. However, I have some concerns and suggestions that should be addressed prior to publication.

Major Concerns

- Your study included several papers on chronic health conditions. However, only one was conducted among people with PD. Why did you choose to explore PD? Why did you explore PD in the title of your study?

- Parkinson's disease symptoms and prognosis are quite different from other diseases explored in your study, for example, patients with PD are more vulnerable to fractures or injuries, but this is not a common prognosis for asthma. I suggest either focusing on a common complication associated with all chronic diseases included, or a discussion on this subject.

- On the topic “Search strategy” provide the date when the initial search for literature started.

- Your most important result is regarding anxiety. Please, provide a better explanation of your analyses for anxiety, why was the analyze was different from depression?

- Did you perform any analysis controlling for the type of chronic disease? Again, since their prognosis is different, I would expect different mental health outcomes.

- In the statement: “Although we specifically looked at common complications in PD, only one of the included interventions was conducted in people with PD [37], indicating that case management interventions in PD are still rare.” Although you have included only one study, it is not clear whether management interventions on PD are rare, or few studies are published in this area.

- I suggest that you cite and include figure 4 in the main text. How this meta-analysis could be directly associated with depressive and anxiety symptoms of Parkinson’s disease considering that you had more studies with other chronic diseases?

6. PLOS authors have the option to publish the peer review history of their article (what does this mean?). If published, this will include your full peer review and any attached files.

Reviewer #1: **Yes: **Dr Khalilah McCants

Reviewer #2: No

Reviewer #3: **Yes: **Randriely Merscher Sobreira de Lima

---

## [Author Response · Author response to Decision Letter 0]

22 Dec 2022

We are thankful for the constructive comments and suggestions of the reviewers. These have most certainly helped to further improve our manuscript. Please find our detailed point-by-point reply and corresponding revisions as file attached.

---

## [Decision Letter · Decision Letter 1]

19 Jan 2023

PONE-D-22-05522R1Case management interventions in chronic disease reduce anxiety and depressive symptoms: A systematic review and meta-analysisPLOS ONE

Dear Dr. Darweesh,

Thank you for submitting your manuscript to PLOS ONE. After careful consideration, we feel that it has merit but does not fully meet PLOS ONE’s publication criteria as it currently stands. Therefore, we invite you to submit a revised version of the manuscript that addresses the points raised during the review process.

We look forward to receiving your revised manuscript.

Kind regards,

Vincenzo De Luca

Academic Editor

PLOS ONE

Journal Requirements:

Reviewers' comments:

Reviewer's Responses to Questions

**Comments to the Author**

1. If the authors have adequately addressed your comments raised in a previous round of review and you feel that this manuscript is now acceptable for publication, you may indicate that here to bypass the “Comments to the Author” section, enter your conflict of interest statement in the “Confidential to Editor” section, and submit your "Accept" recommendation.

Reviewer #2: All comments have been addressed

Reviewer #3: All comments have been addressed

2. Is the manuscript technically sound, and do the data support the conclusions?

Reviewer #2: Yes

Reviewer #3: Yes

3. Has the statistical analysis been performed appropriately and rigorously? 

Reviewer #2: I Don't Know

Reviewer #3: Yes

4. Have the authors made all data underlying the findings in their manuscript fully available?

Reviewer #2: No

Reviewer #3: Yes

5. Is the manuscript presented in an intelligible fashion and written in standard English?

Reviewer #2: Yes

Reviewer #3: Yes

6. Review Comments to the Author

Reviewer #2: The author's edits have significantly improved the story and addressed comments appropriately.

There are some additional minor comments based on the current version of the manuscript:

In the introduction:

"predominantly mood disorders such as depression or anxiety"

It sounded like anxiety was described as a mood disorder, anxiety disorder is not a mood disorder.

Also, perhaps clarify the word anxiety in the edited introduction -- e.g., anxiety symptoms or anxiety disorder? Reading the manuscript edits suggest the authors are referring to symptoms, not disorder.

The word anxiety was also used in the new abstract "depressive symptoms and anxiety". I am not sure if this need to be edited. As long as a clarification exists it could be fine to use the term "anxiety" later on.

It is great the authors included more details about the variable used as covariates.

Perhaps add a sentence specifying the coding, e.g., Age is assumed to be a continuous spectrum but sex is usually binary coded.

It seems to be implied that a lot of the variables used without specific mentioning them as categorical variables were binary and categorical variables coded such as 1 = yes 0 = no, but it's not clear if it's the case. If it's the case, then how are the missing data treated?

The data availability statement seems to indicate the data will be openly accessible, is the included table sufficient for replicating the analysis or there are other raw data suitable for analysis available?

Minor grammar issue on new edits:

e.g., the sentence is running long and may need a transition before "we could not"

"Furthermore, the use of antidepressant or anti-anxiety medication may have affected the results, but since some of the original studies included in this meta-analysis did not report sufficient information on medication use we could not assess this in the metaregression analysis. "

Minor comment on the justification of anxiety analysis with a reduced number of covariates. The reasoning is ok, however, it's unclear why certain covariates are removed and why removing X number of covariates. e.g., perhaps the model won't converge with more covariates or add other justification for selected removal.

Another comment based on the new edit is that now it's clear the literature search was done in 2019, it's been more than two years old. Sometimes it's a good idea to check if the literature changed recently. However, I understand to include a refresh of the literature search would delay publication and it would be an editorial decision.

Reviewer #3: The manuscript by Geerlings et al. has greatly improved after adequately addressing the comments raised in the review process. The authors have addressed all my comments and I recommend the acceptance of the manuscript.

7. PLOS authors have the option to publish the peer review history of their article (what does this mean?). If published, this will include your full peer review and any attached files.

Reviewer #2: **Yes: **Yang Liu

Reviewer #3: **Yes: **Randriely Merscher Sobreira de Lima

---

## [Author Response · Author response to Decision Letter 1]

2 Feb 2023

Please find attached the response to the reviewers.

---

## [Editor Report · Decision Letter 2]

20 Feb 2023

Case management interventions in chronic disease reduce anxiety and depressive symptoms: A systematic review and meta-analysis

PONE-D-22-05522R2

Dear Dr. Darweesh,

We’re pleased to inform you that your manuscript has been judged scientifically suitable for publication and will be formally accepted for publication once it meets all outstanding technical requirements.

Kind regards,

Vincenzo De Luca

Academic Editor

PLOS ONE
---

## [Editor Report · Acceptance letter]

23 Feb 2023

PONE-D-22-05522R2 

 Case management interventions in chronic disease reduce anxiety and depressive symptoms: A systematic review and meta-analysis 

Dear Dr. Darweesh:

I'm pleased to inform you that your manuscript has been deemed suitable for publication in PLOS ONE. Congratulations! Your manuscript is now with our production department. 

Kind regards, 

on behalf of

Dr. Vincenzo De Luca 

Academic Editor

PLOS ONE